# DiT-LSTM-SVAR Model For Portfolios

## Abstract

This paper proposes a novel combined model named DiT-LSTM-SVAR, which successfully integrates time series and the Efficient Markets Hypothesis. This is the first to combine the microstructure of financial markets with deep learning networks to improve the performance of portfolios. We employ the DiT model to predict the upside and downside movements and an information decomposition model based on the SVAR model to identify random walk stocks. The DiT module significantly improves the Matthews correlation coefficient by almost 3%. The annual return of the portfolio is improved by almost 20%. The SVAR module greatly improves the Matthews correlation coefficient by almost 4%. Portfolios constructed using the DiT-LSTM-SVAR module based on market and public information outperformed those created with the DiT-LSTM model. The annual cumulative return of the portfolio is 266.60% and a Sharpe ratio of 1.8.

## 1 Introduction

Financial markets serve as an example of time series forecasting paradigms. Despite some progress in the field of time series algorithms, these studies often overlook the unique characteristics of financial markets when constructing investment portfolios (Liu et al., 2022; Mandi and Guns, 2020; Lin et al., 2022; Zhang et al., 2020). The Efficient Market Hypothesis posits that prices reflect all available information in a strong efficient market, making it extremely challenging to predict future price movements (Fama, 1970). This raises a critical question: Can we honestly and accurately forecast financial market trends, or are we merely attempting to identify latent patterns that may not exist or cannot be fully captured? If such patterns do not actually exist or cannot be thoroughly modeled, forcing such modeling efforts can lead to inaccurate predictions, overfitting, and poor performance of the models on actual market data. This, in turn, can harm investors by providing misleading information and suboptimal investment decisions. How do we combine the microstructure of financial markets with deep learning networks to improve the performance of portfolios?

This paper presents a novel algorithm, named DiT-LSTM-SVAR, for constructing investment portfolios. We demonstrate how the approach enables portfolio construction while ensuring rigorous empirical guarantees. The predictive model integrates three key components: (a) Diffusion Transformer (DiT) for data augmentation, (b) Long Short-Term Memory (LSTM) for forecasting stock returns, and (c) Structural Vector Autoregression (SVAR) for assessing the pricing efficiency of stocks, indicating whether they are random walk. Utilizing data from CSI300 in China, we empirically demonstrate how DiT performs well compared to the simple LSTM model. Furthermore, we construct the portfolio based on the DiT-LSTM-SVAR model demonstrate the necessity of employing the SVAR model in filtering random walk stocks.

Our paper contributes to existing literature in several ways. First, to the knowledge, this paper is the first to integrate the microstructure of financial markets with deep learning, thereby building a bridge between them (Mandi and Guns, 2020; Zhang et al., 2020). We use an information decomposition model based on the SVAR model to identify random walk stocks. We categorize price shocks into market information, public information, private information, and noise. DiT-LSTM-VAR model effectively extends existing time series techniques and portfolio construction from a theoretical perspective (Xiang et al., 2022; Chen et al., 2021; Sun et al., 2024; Shi et al., 2024; Yu et al., 2019).

Secondly, this is the first time the DiT model has been applied to predict the upside and downside movements of high-frequency stock market prices, thus enriching the literature on time-series prediction and feature enhancement. We use the DiT model to enhance financial market data by reverse sampling. Portfolios constructed using the DiT-LSTM model outperformed those created with the simple LSTM model. The addition of the DiT module greatly improves the Matthews correlation coefficient by almost 3%. The annual return of the portfolio is at least 20% higher than the latter.

Third, we demonstrate that selecting stocks with weak-form efficiency with the SVAR module can effectively enhance the cumulative equity of investment portfolios with improved Sharpe ratio, reduced volatility, and maximum drawdowns. We find that stocks with a high proportion of market information are predictable; those with a high proportion of public information exhibit some predictability, while stocks with higher proportions of private information and noise are random walks. The addition of the SVAR module greatly improves the Matthews correlation coefficient by almost 4%. Portfolios constructed using the DiT-LSTM-SVAR module based on market and public information outperformed those created with the DiT-LSTM model. The annual cumulative return of the portfolio is 266.60% and a Sharpe ratio of 1.8.

The rest of the paper is structured as follows. We start with related work in Section 2. Section 3 provides a formal definition of DiT-LSTM-SVAR model. Section 4 displays the experiments results. Section 5 concludes the paper with a discussion of the methods and possible future directions.

## 2 RELATED WORK

**Denoising diffusion probabilistic models (DDPMs).** In recent years, DDPMs have made significant advancements in the field of generative models (Song et al., 2020), particularly in image generation (Tang et al., 2023; Jinhui et al., 2023). They have become an effective method, often outperforming previous Generative Adversarial Networks (GANs) (Goodfellow et al., 2020) in many cases. Recent improvements in DDPMs over the past two years have primarily been driven by optimized sampling techniques (Ho et al., 2020). For example, the introduction of more efficient sampling algorithms has enhanced the model's ability to generate high-quality samples (Ho et al., 2020). Classifier-free guidance has also been widely recognized as a technique that significantly improves sample quality (Nichol et al., 2021; Ho and Salimans, 2022; Baykal et al., 2024). This method enhances the flexibility and adaptability of generative models by combining the capabilities of conditional and unconditional generation models. Within this framework, the model can dynamically adjust the influence of conditional information during the generation process, thereby producing high-quality samples that better meet expectations. In terms of architecture selection, Convolutional U-Nets (Falck et al., 2022) and Transformer (Peebles and Xie, 2023; Bao et al., 2023) architectures are currently the primary choices. U-Net networks have become the default choice for DDPMs due to their effectiveness in handling image generation tasks. Meanwhile, attention-based Transformer architectures have also been introduced and have shown excellent performance. These architectural improvements not only enhance the model's generative capabilities but also excel in diversity and detail handling. In summary, through continuous innovations in sampling techniques, classifier-free guidance methods, and architectural optimizations, DDPMs have shown increasingly outstanding performance in the field of generative models, providing powerful tools for high-quality image generation. This paper applies these techniques to high-frequency stock data to improve stock prediction capabilities.

**Stock Prediction.** While literature has made some progress in stock price prediction, it primarily focuses on the algorithm of sequence forecasting techniques, with insufficient emphasis on financial data characteristics. The literature has made some progress in gradient computation(Mandi and Guns, 2020), loss functions, as well as resampling techniques and overfitting(Zhang et al., 2020). For signal-to-noise ratio and survivorship bias problem,Liu et al. (2022) putting forward the FinRL-Meta open library based on reinforcement learning.

In terms of portfolio construction, the literature is based on predicting the probability of directly picking the stocks with the highest probability of constructing a portfolio (Xiang et al., 2022; Chen et al., 2021), and reinforcement learning strategies (Sun et al., 2024; Shi et al., 2024; Yu et al., 2019) . However, according to the efficient market hypothesis, only stocks consistent with weak effectiveness are not randomly walking. If stocks are inherently randomly walking, their predictions may be only by chance and have weak applicability. Therefore, it becomes a critical task to prioritize the identification of these weakly effective stocks.

The stock price has theoretical efficient components and the noise created by market inefficiencies or information asymmetry(Hendershott and Menkveld, 2014). The stationary components(efficient components) of the stock price is determined by all publicly available information, which also includes private information, since it can be inferred(Hasbrouck, 2015; Albert J Menkveld and Lucas, 2007) .Brogaard et al. (2022) propose that the stationary component includes market-wide information, firm-specific information revealed through trading on private information, and firm-specific information revealed through public news.Kerssenfischer and Schmeling (2024) explore that 15%–35% of the variance in stock returns can be attributed to the release of macro news.

Therefore, we stress that if noise predominates the stock price movements, then the stock is random walk. Even if such stock movements are predicted, there is a risk of model collapse. However, the literature does not consider these factors.

## 3 DiT-LSTM-SVAR Model

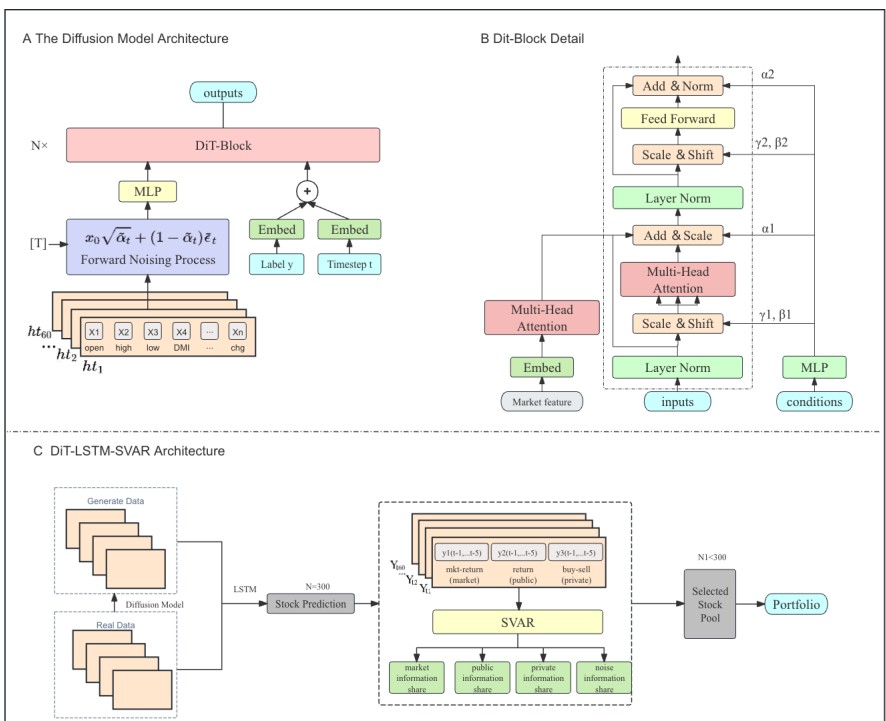

Figure 1: DiT-LSTM-SVAR model.

### 3.1 Preliminaries

**Diffusion Models.** A diffusion model is a type of generative model that generates new data by progressively adding noise to the data and denoising it in the reverse process. The core

idea is to model the data distribution to generate high-quality new samples. The working principle of diffusion models can be divided into two stages:

a) Forward Diffusion Process: The model progressively adds Gaussian noise to the real data $x_0$, making it increasingly random. This process usually consists of a series of time steps, from the initial data distribution to the final pure noise distribution 1.

b) Reverse Denoising Process: Through learning a denoising network, the forward diffusion process is reversed, gradually restoring the original data from pure noise. This process is also conducted step by step, with each step generating data dependent on the denoising result of the previous step $p_\theta(x_{t-1}|x_t) = \mathcal{N}\left(\mu_{\theta(x_t)}, \sum_\theta(x_t)\right)$.

**Classifier-Free Guidance.** It is a technique used in generative models, particularly in diffusion models, aimed at improving the quality and diversity of generated samples without the need for a separate classifier model. Within the classifier-free guidance framework, the reverse process becomes $p_\theta(x_{t-1} \mid x_t, c)$. Classifier-free guidance is widely recognized for producing significantly superior samples compared to conventional sampling techniques . Therefore, adopting classifier-free guidance for generating high-frequency stock data is highly suitable. It not only captures the complex patterns and variations in the data but also enhances the model's sensitivity to market fluctuations and the accuracy of predictions.

## 3.2  Diffusion Transformer Design

Inspired by the diffusion model architecture proposed by Peebles and Xie (2023), we have developed a new DiT-LSTM-SVAR model. This model introduces market feature and conditional inputs based on DiT, aiming to enhance stock prediction data. By integrating these new elements, our goal is to generate more similarly distributed high-frequency stock data within the DiT framework to improve the accuracy and reliability of predictions. Figure 1 presents an overview of the complete XXX architecture, detailing the various components of the model and their interrelationships.

**Forward Noising Process.** The Forward Noising Process is a crucial component of diffusion models, which progressively transforms data into noise. In the Forward Noising Process, Gaussian noise is incrementally added to the data over a series of time steps. The goal is to transform the original data distribution into a noise distribution, typically a Gaussian distribution. The process is usually formulated as follows:

$$q = (x_t|x_0) = \mathcal{N}\left(x_t; \sqrt{\alpha}x_0, (1 - \sqrt{\alpha})\boldsymbol{I}\right) \tag{1}$$

**Dit-Block Design.** After the Forward Noising Process, the output noise data first undergoes spatial dimensional changes through an MLP, and then is processed by a series of transformer blocks. In addition to the noisy high-frequency stock data inputs, the diffusion model also integrates return and timestamp encoding information. Details are shown in Figure 1(a).

*Market factors* Market factors include stock index futures returns. By integrating these market factors, we enhance the performance of portfolios. Stock index futures prices can reflect investors' expectations of future market trends and provide additional information about market movements and volatility. Within the framework of the diffusion model, we input market factors along with high-frequency stock data and timestamp encoding information into the DiT block, utilizing a multi-head attention mechanism to process this information. The multi-head attention mechanism assigns weights to these inputs, effectively reducing the risk of model collapse by giving appropriate weight to the market factors module. This integration through the multi-head attention mechanism allows the model to better understand market dynamics by combining market factors with high-frequency stock data and timestamp encoding. Ultimately, this enables the model to capture subtle changes in high-frequency data while leveraging market factors to improve overall predictive performance.

*conditions.* We embed the timestamp $t$ and the vector of high-frequency stock data fluctuations $y$, and then add them together as additional conditions. This approach not only helps the model capture temporal changes and the volatility of high-frequency stock data, but also enhances the understanding and utilization of this information for more accurate predictions.

By combining timestamp and fluctuation data, the model demonstrates greater robustness and predictive performance in more complex market environments. The specific implementation is shown in Figure 1, illustrating how these additional conditions are integrated into our model architecture.

### 3.3 PREDICTION MODULE DESIGN

In our prediction model, we used LSTM and GRU, and trained them separately with augmented data to test the effects of data augmentation. This approach allows us to evaluate the performance of data augmentation on different models.

### 3.4 SELECTING MODULE DESIGN

We use the SVAR model to calculate the pricing efficiency of stocks. We assume that stock information comes from market information, public information, private information, and noise. Market information refers to information related to the entire market, reflecting the overall market trend and expectations. We use the market portfolio return as its proxy variable. Public information refers to company information disclosed to the public, and we use historical returns as its proxy variable. Private information refers to information possessed by only a few specific individuals or groups, and we use the buy-sell imbalance indicator as its proxy variable. If noise information is too large, indicating that stocks follow a random walk, then prediction becomes difficult.

Analyzing pricing efficiency is rooted in the state-space model (Hendershott and Menkveld, 2014; Hasbrouck, 2015; Albert J Menkveld and Lucas, 2007; Brogaard et al., 2022).

$$p_t = m_t + s_t, \tag{2}$$
$$m_t = m_{t-1} + \mu + w_t, \tag{3}$$
$$w_t = \theta_0 \epsilon_{m,t} + \theta_r \epsilon_{r,t} + \theta_x \epsilon_{x,t}, \tag{4}$$

where $p_t, m_t, \mu, w_t, s_t$ are observed prices, true values, the discount rate, random-walk innovations, and pricing errors at time $t$. $\mathbb{E}_t[w_t] = 0$. $\epsilon_{m,t}, \epsilon_{r,t}$, and $\epsilon_{x,t}$ represent unexpected innovation in market return. The Bitcoin return:

$$r_t = p_t - p_{t-1} = \mu + \theta_r \epsilon_{r,t} + \theta_x \epsilon_{x,t} + \Delta s_t. \tag{5}$$

It includes: the discount rate $\mu$, the components of the efficient market hypothesis (the innovations $w_t$), and pricing errors ($\Delta s_t$). Information variances:

$$\sigma_r^2 = \theta_r^2 \sigma_{\epsilon_r}^2 + \theta_x^2 \sigma_{\epsilon_x}^2. \tag{6}$$

By normalizing these variance components to sum to 100%, we get the shares of variance attributable to different components. See Appendix B for details.

$$\text{Noise Share} = \frac{\sigma_{\epsilon_x}^2}{\sigma_w^2 + \sigma_{\epsilon_x}^2}, \quad \text{Market Info Share} = \frac{\sigma_r^2}{\sigma_w^2 + \sigma_{\epsilon_x}^2},$$
$$\text{Public Info Share} = \frac{\theta_r^2 \sigma_{\epsilon_r}^2}{\sigma_w^2 + \sigma_{\epsilon_x}^2}, \quad \text{Private Info Share} = \frac{\theta_x^2 \sigma_{\epsilon_x}^2}{\sigma_w^2 + \sigma_{\epsilon_x}^2}. \tag{7}$$

We use reduced-form VAR to estimate the SVAR model.

### 3.5 PORTFOLIO MODULE DESIGN

After utilizing the DiT-LSTM model to forecast the probability of stock price fluctuations, we opt for the ten stocks with the highest likelihood of upward movement to form an investment portfolio. Initially, we set thresholds for private, public, market, and noise-related information, thereby eliminating stocks deemed challenging to predict. Subsequently, we contrast the investment portfolios crafted after removing these unpredictable stocks with those directly derived from the DiT-LSTM and LSTM models' predictions of stock price movements. We evaluate various metrics, including cumulative holding period returns, standard deviation, realized volatility, maximum drawdown, and Sharpe ratio. Throughout the portfolio construction process, we employ both equal-weighting and minimum variance strategies based on historical returns and covariance matrices.

## 4 EXPERIMENTS

### 4.1 DATASETS

Our dataset consists of 300 stocks from the CSI300, at hourly frequency spanning from 2022-01-04 to 2024-04-30, with the testing set covering the period from 2024-01-02 to 2024-04-30. Seven fundamental features include opening price, closing price, highest price, lowest price, volume, amount, and price change. Nine technical features include BIAS, BOLL, DMI, EXPMA, HV, KDJ, MA, MACD, and RSI. The specific explanations for each technical indicator are provided in the Appendix A. To assess the influence of the market factor on the generation process, we respectively select the returns of Chinese CSI300 stock index futures. For each stock at hour t, we look back 60 hours to construct a sequence of factor as $R^{60 \times 16}$.

For SVAR selecting model, We used buy and sell volume data to construct a buy-sell imbalance indicator. We collected historical market portfolio returns, buy-sell imbalance indicators, and corresponding stock returns. Each SVAR model used a sample interval of the past 60 hours. Subsequently, we calculated the pricing efficiency for each stock on a rolling basis, determining the proportion of market information, private information, public information, and noise.

### 4.2 BASELINES

**GRU**: This model trains a forecasting framework comprising a GRU layer with a hidden size of 32, a linear layer, and an activation layer using Sigmoid.

**LSTM**: This model trains a forecasting framework comprising an LSTM layer with a hidden size of 32 and a linear layer, incorporating market factors.

**DiT-LSTM**: This approach first uses DiT model to generate additional data according to opposite labels to balance samples, and then trains a forecasting framework comprising an LSTM layer with a hidden size of 32 and a linear layer. We also test the case when using DiT model to generate additional data according to same labels and then add it to the training set with a probability of 0.2.

**DiT-LSTM-SVAR**: This approach first uses a DiT block to generate additional data and then filter random walk stocks. We filter random walk stocks with market and public information, market information, and noise information, respectively, based on SVAR model. See Appendix B for details.

### 4.3 EXPERIMENTAL SETUP

**Software and Hardware**: DiT-LSTM-SVAR is implemented with Python 3.8, Pytorch 1.11.0. We run the experiments on servers equipped with RTX 3080(10GB) GPU and Intel(R) Xeon(R) Platinum 8255C CPU @ 2.50GHz CPU.

**DiT Module**: We frame the task as a binary classification problem based on the stock returns. During the training process, we augmented the original DiT model by introducing a cross-attention mechanism to explore the impact of information from the Chinese markets on the data generation process. We incorporated this mechanism by integrating it with the original features using a weighted fusion approach, weighting at 0.5. We set the learning rate of the DiT model to 1e-5 and the learning rate of the downstream classification model to 5e-4. In the generation process, we adopt a classifier-free guidance approach for sampling.

**Portfolio**: In calculating the cumulative return on investment, we follow simple assumptions: (1) Traders filter the random stock.(2) Traders spend the same amount on each trading day. (3) There is always enough liquidity in the market. (4)Transaction costs are ignored.

**Evaluation Metrics for prediction**: (1) Accuracy: Accuracy is the proportion of correctly predicted instances out of the total instances. For imbalanced data, it cannot fully capture the performance. (2) F1 Score: The F1 score is the harmonic mean of precision and recall, providing a balance between these two metrics. It considers both false positives and false negatives in the evaluation. (3) MCC: Matthews Correlation Coefficient(MCC) is computed

based on all four elements of the confusion matrix (true class, false positive class, true negative class, false negative class), providing a balanced performance evaluation, even in datasets with extremely unbalanced classes.

**Evaluation Metrics for Portfolio**: (1) Cumulative Return: Cumulative return is the total return on an investment over a period of time. (2) Annual Return: Annualized rate of return is the conversion of an investment's total return to an annual percentage. (3) MDD: Maximum drawdown measures the maximum loss of a portfolio from peak to trough over a selected time period. It is an important measure of downside investment risk. (4) Annualized STD: Annualized standard deviation is a measure of the volatility or risk of investment returns. (5) Sharpe: The Sharpe ratio is a measure of an investment's return relative to its risk. A high Sharpe Ratio indicates a higher return on investment per unit of risk assumed. (6) Sortino: Sortino ratio, similar to the Sharpe ratio, only takes into account downside risk rather than overall standard deviation. (7) VaR95: Value at Risk quantifies the probability of the largest possible loss in the value of an investment over a given period of time with 95% confidence. see Append B.1 for details.

## 4.4 Results

**Prediction**.Table 1 shows the accuracy and F1 score of different models. Our sample labels are imbalanced, which is evident in both the training and testing datasets. Hence, we use a DiT model to balance the labels. If the stock has a downside movement, we label them 1, otherwise 0. In the DiT module sampling process, P stands for normal sampling and N uses negative sampling. MP, M, and M stand for use represent the use of market and public information filtering, market information filtering, and noise filtering.

Table 1: Performance comparison of different models (%)

| Market | GRU | LSTM | DiT -LSTM(P) | DiT -LSTM(N) | DiT-LSTM -SVAR(MP) | DiT-LSTM -SVAR(M) | DiT-LSTM -SVAR(N) |
|---|---|---|---|---|---|---|---|
| ACC | 50.70 | 54.04 | 54.82 | 54.57 | 54.39 | 52.01 | 54.68 |
| F1 | 58.07 | 69.16 | 69.44 | 69.70 | 69.54 | 66.98 | 69.80 |
| MCC | -0.92 | 0.89 | 4.35 | 3.07 | 4.25 | 7.54 | 3.08 |
| No Market | | | | | | | |
| ACC | 52.79 | 54.31 | 54.74 | 55.15 | 55.10 | 53.44 | 55.30 |
| F1 | 65.32 | 70.39 | 69.37 | 66.14 | 69.05 | 66.13 | 69.23 |
| MCC | -0.11 | 0.00 | 4.04 | 5.66 | 6.81 | 9.48 | 5.81 |

With the market factor, the DiT module significantly improves the prediction. According to the baseline, the GRU model has an accuracy of 50.70%, an F1 score of 58.07% and an MC of -0.92%. It shows that the model has no predictive ability. The LSTM model has an accuracy of 54.04%, F1 score of 69.16% and MC of 0.89%. The GRU and LSTM predict the vast majority of labels as falling, which happens to be a larger proportion of falling labels in the test set. The DiT module significantly improves the prediction. After sampling with the DiT module, the accuracy reached 54.82% and 54.57%, F1 scores reached 69.44% and 69.70%, and MC reached 4.35% and 3.07%, with and without negative sampling respectively. The model exhibits higher F1 scores and effectively reduces single label prediction. The SVAR module further improves the prediction. The MC further reaches 7.54% after filtering random wandering stocks using market information. As the MC integrates the four cases of the confusion matrix, it indicates that the generalization ability of the model is further improved, effectively improving the single-label prediction.

Without market factor, the DiT also module significantly improves the prediction. For baseline, the GRU model has an accuracy of 52.79%, an F1 score of 65.32% and an MC of -0.11%. It also displays that the model has no predictive ability. The LSTM model has an accuracy of 54.74%, F1 score of 68.97% and MC of 4.17%. After sampling with the DiT module, the accuracy reached 54.74 % and 55.15%, F1 scores reached 69.37% and 66.14%, and MC reached 4.04% and 5.66%, with and without negative sampling respectively. Negative sampling performs well with a higher MC. The SVAR module further improves the

prediction. The MC further reaches 9.48% after filtering random wandering stocks using market information. The generalization ability of the model is further improved.

Although the DiT-LSTM-SVAR model improves prediction ability, the prediction of LSTM itself could be better. Subsequent experiments need to improve the LSTM model, such as using the XLSTM model.

**Portfolio**. Figure 2 present the portfolio performance. Overall, both the CSI300 and LSTM models perform poorly, with cumulative equity of approximately 1.05 and 1.13, respectively. In contrast, the DiT-LSTM and DiT-LSTM-SVAR models significantly outperform the CSI300 and LSTM models when processing various types of information, including market, market and public, market, public and private (the sum is 1−noise). Among them, the DiT-LSTM-SVAR model performs the best in all scenarios, especially when combining public and market information shares, achieving the highest cumulative equity of 1.46.

With the market factor, the cumulative equity of the portfolios constructed using the DiT-LSTM-SVAR model based on both market and public information, is 1.46, which is significantly higher than the DiT-LSTM model's 1.18. The cumulative equity of the portfolios constructed based on market information is 1.36, and based on the noise information is 1.24. The latter two are also higher than the DiT-LSTM model's 1.18.

Without the market factor, the cumulative equity of the portfolios constructed using the DiT-LSTM-SVAR model based on both market and public information, is 1.35, which is significantly higher than the DiT-LSTM model's 1.23. The cumulative equity of the portfolios constructed based on market information is 1.33, and based on the noise information is 1.26. The latter two are also higher than the DiT-LSTM model's 1.23.

Moreover, the DiT-LSTM-SVAR model, incorporating financial characteristics, significantly enhances the model's generalization ability. In the latter stages of the test sample, the performance of portfolios constructed by the LSTM and DiT-LSTM models declined, possibly due to macroeconomic changes. However, after integrating the SVAR model with financial characteristics, the portfolio's performance remained stable, avoiding overfitting issues solely reliant on time series prediction techniques and significantly reducing the portfolio's volatility and maximum drawdown.These results demonstrate the significant value of the DiT-LSTM-SVAR model in stock prediction and investment strategy, outperforming the DiT-LSTM model, especially when various types of information are combined to significantly enhance investment returns.

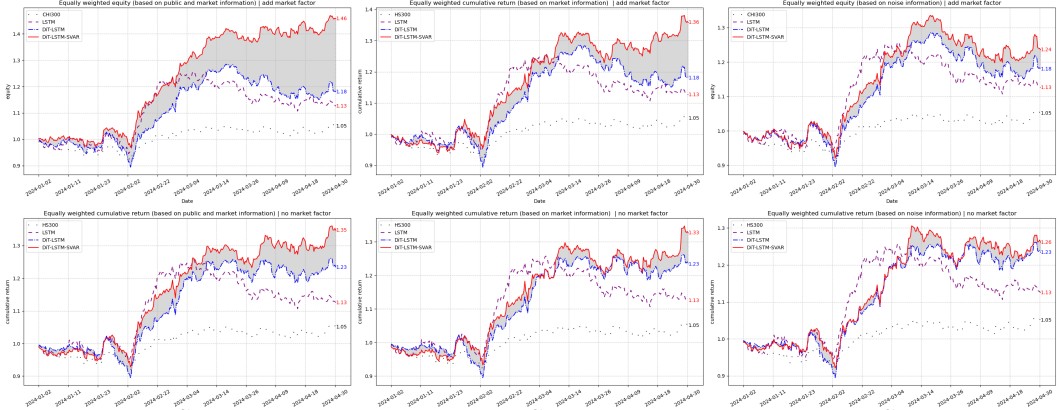

Figure 2: Portfolio performance

Table 2 shows the performance of portfolios constructed with the CSI 300, the LSTM model, the DiT-LSTM model with and without market factors, and the DiT-LSTM-SVAR model combined with the DiT-LSTM-SVAR model with filter random-walk stocks. We chose to select an equal number of 10 stocks. We change the time series scale and portfolio strategy in the AppendixE and the results remain robust. The results show that relying only on the LSTM model for portfolio selection is not effective. Although the annualized return can be

Table 2: Portfolio performance comparison

| Model | Cumulative Return | Annual Return | MDD | Annual STD | Sortino | VaR95 | Sharpe |
|---|---|---|---|---|---|---|---|
| CHI300 | 5.05% | 18.54% | -7.39% | 16.04% | 0.52 | -0.86% | 0.31 |
| LSTM | 12.58% | 50.55% | -12.04% | 26.21% | 0.82 | -1.25% | 0.48 |
| DiT-LSTM (Market factor) | 17.97% | 76.89% | -12.87% | 28.30% | 1.14 | -1.19% | 0.63 |
| DiT-LSTM(no market factor) | 23.35% | 106.35% | -12.46% | 27.54% | 1.36 | -1.35% | 0.85 |
| DiT-LSTM-SVAR(MP) | 45.69% | 266.60% | -7.10% | 25.40% | 3.41 | -1.01% | 1.80 |
| DiT-LSTM-SVAR(M) | 35.79% | 187.49% | -8.74% | 28.52% | 2.46 | -1.21% | 1.25 |
| DiT-LSTM-SVAR(N) | 23.55% | 107.49% | -10.42% | 28.50% | 1.47 | -1.21% | 0.83 |
| DiT-LSTM-SVAR(no market factor,MP) | 34.64% | 179.18% | -9.56% | 26.79% | 2.21 | -1.26% | 1.29 |
| DiT-LSTM-SVAR(no market factor, M) | 32.64% | 165.11% | -8.59% | 29.24% | 2.18 | -1.25% | 1.12 |
| DiT-LSTM-SVAR(no market factor, N) | 25.70% | 120.26% | -12.50% | 27.84% | 1.53 | -1.38% | 0.92 |

up to 50.55%, it is significantly lower than the other models.The Sortino and Sharpe ratios are only 0.82 and 0.48.The maximum backtest is -12.04% Using DiT Balanced Labeling, the data enhancement resulted in an increase in the effectiveness of the portfolios selected by the model. The annualized return rises to 76.89%, but is significantly lower than the portfolio using the SVAR model.Sortino and Sharpe improve to only 1.14 and 0.63. Subsequently, we used the SVAR model to filter random wandering stocks. The portfolio constructed by the DiT-LSTM-SVAR model with market factors and based on market and public information performs the best with an annualized return of 266.60% and a volatility of 25.40%. the Sortino and Sharpe ratios reach 3.41 and 1.80 and the maximum retracement is only 7.10%. The portfolio constructed by the DiT-LSTM-SVAR model with market factors and based on publicly available information can achieve an annualized return of 197.49% with a volatility of 28.52%, Sortino and Sharpe ratios of 2.46 and 1.25, and a maximum retracement of 8.74. However, the DiT-LSTM-SVAR model based on noise does not improve much. Overall, portfolios constructed with the inclusion of market factor forecasts perform better than those constructed without them. In the case of fliter stocks based on market information and public information, the portfolio constructed by adding the market factor has an annualized return of 266.60%, which is significantly higher than the portfolio constructed without the market factor, which is 179.18. The portfolio constructed with the market factor also outperforms the portfolio constructed without the market factor in the case of fliter stocks based on public information and noise.

Our portfolio outperforms the existing literature. After filtering stochastic stocks using SVAR model with financial characteristics, our portfolio has an annualized return of up to 266.60%, Sharpe ratio of up to 1.80, Sortino ratio of up to 3.41, and a maximum drawdown of only 7.10%. (Xiang et al., 2022) also construct portfolios based on CSI300 . Although their Sharpe ratio is 1.88, the annualized return is 63.2%, the maximum drawdown is 23.7%. (Xiang et al., 2022) construct portfolios based on CSI500, with a return of 235.3%, a Sharpe ratio of 1.4. (Shi et al., 2024)'s portfolio based on China ChiNext has a return of 31.1%, a Sharpe ratios of 0.907, and a Sortino ratio of 1.236. (Chen et al., 2021)'s portfolio based on Chinese sector has a annualized return of 214%, a Sharpe Ratio approximation of 1.8, a maximum drawdown of 40%. In the U.S. market, (Yu et al., 2019)'s portfolio has an annualized return of 8.09%, a Sharpe ratio of 0.63, and a Sortino ratio of 0.89. (Sun et al., 2024)'s portfolio has a return of 57.15%, a Sharpe ratio of 0.804, and a Sortino ratio of 1.564.

## 5 CONCLUSION

We construct the DiT-LSTM-SVAR to facilitate the development of the time series domain, which will have a broad impact on academic researchers and financial practitioners. The DiT-LSTM-SVAR model is a diffusion-based deep learning prediction model combined with an SVAR-based portfolio selection model. It eliminates random-walk stocks to construct robust, high-return investment portfolios. We predict the price movements of CSI 300 stocks and constructed investment portfolios by filtering out random walk stocks using market, public, private, and noise information shares. Given that constructing portfolios aims to pick the stocks with the highest returns efficiently, we introduce the concept of effective forecasting accuracy. We demonstrated the necessity of filtering random walk stocks by comparing the performance of portfolios constructed after the SVAR filtering model and portfolios directly constructed using the DiT-LSTM model. We also show that incorporating

the DiT module enhances the data by balancing the sample labels, which can dramatically increase recall and pinpoint rising stocks.

This paper theoretically bridges the fields of time series analysis and financial data. We use an information decomposition model based on the SVAR model to identify random walk stocks. By integrating the SVAR model into the time series forecasting technique, our model can be better applied to portfolio construction based on time series forecasting. This effectively complements the previous literature that did not fully consider portfolio construction and focused on stock price series forecasting(Liu et al., 2022; Mandi and Guns, 2020; Lin et al., 2022; Zhang et al., 2020).

Since market information affects stock prices, we add market factors to the forecasting model. Our research findings have significant implications for portfolio construction. We find that selecting stocks with strong explanatory power in private and public information can significantly improve portfolio performance.A portfolio constructed using the DiT-LSTM-SVAR model can have up to 0.28 more holding equity than a portfolio constructed using the DiT-LSTM model. Even without adding the market factor, the portfolios constructed based on DiT-LSTM-SVAR outperform the DiT-LSTM model. Our portfolios have higher annualized cumulative returns, higher Sharpe ratios, and lower maximum drawdown than the existing literature(Yu et al., 2019; Shi et al., 2024; Sun et al., 2024).These results underscore the practical value of our model in enhancing portfolio performance. Our DiT-LSTM-SVAR-based model demonstrates a high level of generalization ability, a key advantage over the LSTM model which often suffers from overfitting. This overfitting can lead to less effective prediction of the later period than the earlier period. By highlighting this superiority, we instill confidence in the effectiveness of our model, further reinforcing its practical relevance and applicability.

As the data is unbalanced due to more downward movements of stocks during the sample period, we include DiT in our forecasting model to augment the data. The results of our study complement the time series forecasting. We find that the addition of the DiT module improves the recall of predicting downside movements by nearly 30%, makes the portfolios more conservative. The cumulative equity of the portfolios constructed using the DiT-LSTM-SVAR model is 0.28 higher than that of the portfolios constructed using the DiT-LSTM model with the market factor, and Sortino rises by 0.32, which effectively reduces the downside risk. These results emphasize the practical value of our DiT model in improving stock forecasting, further enhancing its practical relevance and applicability, especially in financial markets where investors are more sensitive to stock declines than stock rises.

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

## A Definition

- **MA(Moving Average):** The MA indicator is utilized to smooth price data for the identification of trend direction and strength. Characterized by its simplicity and intuitive nature, it represents the average of prices over a specified historical period. The formula is as follows:

$$MA = \frac{Close_1 + \ldots + Close_n}{n} \tag{A1}$$

Here we set $n$ to be 5.

- **EXPMA(Moving Average):** The EXPMA indicator is characterized by an exponentially decreasing weighted moving average. Its construction principle entails calculating the arithmetic mean of stock closing prices and utilizing the resultant values for analysis, aiding in the assessment of the future directional changes in price trends.

- **MACD(Moving Average Convergence Divergence):** The MACD indicator represents a progressive extension of the moving average principle. The computational procedure is delineated as follows:

  1. Fast Line Calculation: The calculation involves determining the difference between the short-term moving average and the long-term moving average. The short-term EMA is computed over a 12-hour period, while the long-term EMA is computed over a 26-hour period.
  2. Slow Line Computation: The slow line is derived by computing an exponential moving average over a period of $n$ days on the fast line. Here, we set $n$ to be 9.
  3. Calculation of MACD Histogram: This involves determining the difference between the fast line and the slow line.

- **BIAS:** BIAS is a commonly used technical indicator in quantitative trading, primarily employed to measure the deviation of prices from the moving average line. Its calculation formula is as follows:

$$BIAS = \frac{Close - MA_n}{MA_n} \tag{A2}$$

Here we set $n$ to be 12.

- **BOLL(Bollinger Bands):** The BOLL indicator is a technical analysis tool based on price volatility, utilized to assess the fluctuation and trend changes of prices. The Bollinger Bands indicator comprises three lines:

  1. Middle Band: Typically, it is a moving average line over a certain period of time, commonly observed as a Simple Moving Average (SMA), Exponential Moving Average (EMA), or other types of moving averages.
  2. Upper Band: Positioned above the middle band, it is separated from the middle band by a certain number of standard deviations. Typically, it is calculated by adding multiples of standard deviations to the middle band.
  3. Lower Band: Positioned below the middle band, it is separated from the middle band by a certain number of standard deviations. Typically, it is calculated by subtracting multiples of standard deviations from the middle band.

- **DMI(Directional Movement Index):** The DMI indicator is primarily employed to gauge the strength of stock price trends and potential reversal points, comprising the Positive Directional Indicator (+DI), Negative Directional Indicator (-DI), and Trend Indicator. In an upward trend, a buy signal is indicated when +DI crosses above -DI, while a sell signal is indicated when +DI crosses below -DI. The trend indicators, namely ADX (Average Directional Index) and ADXR (Average Directional Movement Rating), guided by +DI and -DI, are pivotal indicators for identifying market trends.

- **HV(Historical Volatility):** The HV indicator is the volatility calculated based on the price changes of a stock over a certain period of time. It reflects the actual volatility of the stock during historical periods.

- **KDJ:** The KDJ indicator is primarily utilized to identify overbought and oversold conditions of stock prices, as well as to recognize reversal points in price trends. Derived from the Stochastic indicator, the KDJ indicator comprises three lines: the K line, the D line, and the J line.

  1. K Line: The K line is the primary curve, representing the relative position of the closing price to the highest and lowest prices over a recent period of time.
  2. D Line: The D line is the moving average of the K line, employed to smooth the fluctuations of the K line. The calculation of the D line typically involves a simple moving average or exponential moving average over a period of time.
  3. J Line: The J line is calculated based on the K line and the D line. It is utilized to gauge overbought and oversold conditions of prices, as well as to identify reversal points in price trends.

- **RSI:** The RSI indicator is a technical tool utilized to measure the velocity and magnitude of price changes in stocks. It is based on comparing the magnitude of price increases and decreases over a certain period, as well as the average magnitude of price changes, to determine overbought and oversold conditions in the market.

## B  Building a Portfolios

After we get the prediton of DiT-LSTM model, We use Structural Vector Autoregression (SVAR) to assess the random walk characteristics of stocks in order to select those with strong predictability. The selection is based on the following criteria lists on Figure B.:

- We choose stocks where the market information share exceeds 0.3.
- We select stocks where the combined share of market and public information is greater than 0.67.
- We pick stocks where the sum of market information share, public information share, and private information share is more than 0.75, implying that the noise share is less than 0.35.

We construct the portfolio using an equal-weight approach. We have selected the top 10 stocks with the highest probability of rising. Assuming an initial equity of 1, we calculate the cumulative equity over time. The final cumulative equity minus the initial equity, divided by the initial equity, yields the cumulative return rate of the portfolio.

We evaluate the performance of our portfolio using several key financial metrics: cumulative return rate, standard deviation, realized volatility, Sharpe ratio, and maximum drawdown. These metrics provide a comprehensive view of the portfolio's performance and risk characteristics, detailed as follows:

### B.1  Formlas

Cumulative Return measures the total increase in value of an investment over a set period of time, expressed as a percentage.

$$CR = \left( \frac{P_{end} - P_{start}}{P_{start}} \right) \times 100\% \tag{B1}$$

where $P_{start}$ is the initial price of the investment and $P_{end}$ is the price of the investment at the end of the period.

Annual Return is the yearly average percentage return realized by an investment. It is standardized to a yearly scale, allowing for comparison across different time frames.

$$AR = (1 + CR)^{\frac{252}{n}} - 1 \tag{B2}$$

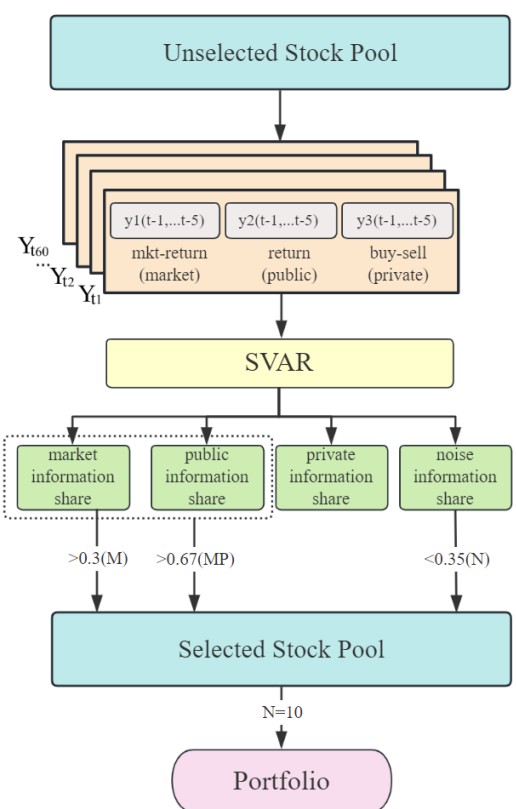

Figure B1: Portfolio Building Process

where $CR$ is the cumulative return and $n$ is the number of test days.

Maximum Drawdown is the maximum observed loss from a peak to a trough of a portfolio, before a new peak is attained. It is usually quoted as a percentage. The formula is:

$$MDD = \min\left(\frac{T_{min} - P_{peak}}{P_{peak}}\right) \tag{B3}$$

where $T_{min}$ is the minimum trough value after the peak $P_{peak}$.

Annualized Standard Deviation measures the volatility of an investment's returns. It is calculated as:

$$\sigma_{annual} = \sigma\sqrt{\frac{252}{n} \times 4} \tag{B4}$$

where $\sigma$ is the standard deviation of returns and $n$ is the number test days. 4 is the number of trading hour in a day.

Value at Risk (VaR) measures the maximum potential loss in value of a risky asset or portfolio over a defined period for a given confidence interval. The 95% confidence level means there is only a 5% chance that the loss will exceed the VaR.

The Sharpe Ratio measures the performance of an investment compared to a risk-free asset, after adjusting for its risk. It is calculated as:

$$Sharpe = \frac{R_p - R_f}{\sigma} \tag{B5}$$

where $R_p$ is the return of the portfolio, $R_f$ is the risk-free rate, and $\sigma$ is the standard deviation of the portfolio's excess return over the risk-free rate.

Sortino Ratio modifies the Sharpe Ratio by differentiating harmful volatility from total overall volatility by using the standard deviation of negative asset returns, called downside

deviation.

$$Sortino = \frac{R_p - R_f}{\sigma_d} \tag{B6}$$

where $R_p$ is the portfolio return, $R_f$ is the risk-free rate, and $\sigma_d$ is the downside deviation. We set $R_f$ zero.

## C SVAR AND PRICING EFFICIENCY

Consider the log of the observed price at time $t$, $p_t$, as the sum of two components(Brogaard et al., 2022):

$$p_t = m_t + s_t, \tag{C1}$$

where $m_t$ is the efficient price, and $s_t$ is the pricing error. The pricing errors can have a temporary (short-run) effect on the price, but they do not affect the price in the long run (no permanent effect). $m_t$ follows a random walk with drift $\mu$ and innovations $w_t$:

$$m_t = m_{t-1} + \mu + w_t. \tag{C2}$$

The innovations reflect new information about the fundamentals and are thus unpredictable:

$$\mathbb{E}_{t-1}[w_t] = 0. \tag{C3}$$

The drift is the discount rate over the next period (day). The return is:

$$r_t = p_t - p_{t-1} = \mu + w_t + \Delta s_t. \tag{C4}$$

We partition the information impounded into three sources: market-wide information, private information incorporated through trading, and public information such as cryptocurrency-related news. This partitioning is essential for understanding the dynamics of pricing and how information is incorporated into prices. The random-walk innovations, $w_t$, can then be decomposed into three parts:

$$w_t = \theta_{im}\epsilon_{im,t} + \theta_x\epsilon_{x,t} + \theta_r\epsilon_{r,t}, \tag{C5}$$

and thus:

$$\begin{aligned}
r_t = p_t - p_{t-1} &= \mu + w_t + \Delta s_t \\
&= \mu + \theta_{rm}\epsilon_{rm,t} + \theta_x\epsilon_{x,t} + \theta_r\epsilon_{r,t} + \Delta s_t \\
&= discount \quad + \underbrace{\theta_{rm}\epsilon_{rm,t}}_{market\ info} \quad + \underbrace{\theta_x\epsilon_{x,t}}_{private\ info} \quad + \underbrace{\theta_r\epsilon_{r,t}}_{public\ info} \quad + \underbrace{\Delta s_t}_{noise}
\end{aligned} \tag{C6}$$

- $\epsilon_{rm,t}$: the unexpected innovation in the market return
- $\theta_{rm}\epsilon_{rm,t}$: the market-wide information incorporated into stock prices
- $\epsilon_{x,t}$: an unexpected innovation in signed dollar volume
- $\theta_x\epsilon_{x,t}$: the firm-specific information revealed through trading on private information
- $\theta_r\epsilon_{r,t}$: the remaining part of firm-specific information that is not captured by trading on private information ($\epsilon_{r,t}$ is the innovation in the stock price).
- Changes in the pricing error, $s_t$, can be correlated with the innovations in the efficient price $w_t$.

Both the permanent (information) and transient (noise) components are driven by the same shocks. The noise is defined as:

$$\Delta s_t = r_t - \mu - w_t = r_t - a_0 - \theta_{rm}\epsilon_{rm,t} - \theta_x\epsilon_{x,t} - \theta_r\epsilon_{r,t} \tag{C7}$$

To understand the variance in the information, we decompose it as follows:

- *Information variance*

$$Eff = \sigma_w^2 = \theta_{rm}^2 \epsilon_{rm}^2 + \theta_x^2 \epsilon_x^2 + \theta_r^2 \epsilon_r^2 \tag{C8}$$

$$Noise = \sigma_s = \text{Var}(\Delta s_t) \tag{C9}$$

$$MktInfo = \theta_{rm}^2 \sigma_{\epsilon_{rm}}^2 \tag{C10}$$

$$PrivateInfo = \theta_x^2 \sigma_{\epsilon_x}^2 \tag{C11}$$

$$PublicInfo = \theta_r^2 \sigma_{\epsilon_r}^2 \tag{C12}$$

By normalizing these variance components to sum to 100%, we get the shares of variance attributable to different components:

$$\text{NoiseShare} = \frac{\sigma_s^2}{\sigma_w^2 + \sigma_s^2}, \quad \text{MarketInfoShare} = \frac{\theta_{im}^2 \sigma_{im}^2}{\sigma_w^2 + \sigma_s^2},$$
$$\text{PublicInfoShare} = \frac{\theta_r^2 \sigma_r^2}{\sigma_w^2 + \sigma_s^2}, \quad \text{PrivateInfoShare} = \frac{\theta_x^2 \sigma_x^2}{\sigma_w^2 + \sigma_s^2}. \tag{C13}$$

Our estimation is based on a structural VAR with five lags.

$$r_{m,t} = \sum_{l=1}^{5} a_{1,l} r_{m,t-l} + \sum_{l=1}^{5} a_{2,l} x_{t-l} + \sum_{l=1}^{5} a_{3,l} r_{r,t-l} + \epsilon_{r_m,t}$$

$$x_t = \sum_{l=0}^{5} b_{1,l} r_{m,t-l} + \sum_{l=1}^{5} b_{2,l} x_{t-l} + \sum_{l=1}^{5} b_{3,l} r_{r,t-l} + \epsilon_{x,t} \tag{C14}$$

$$r_t = \sum_{l=0}^{5} c_{1,l} r_{m,t-l} + \sum_{l=1}^{5} c_{2,l} x_{t-l} + \sum_{l=1}^{5} c_{3,l} r_{r,t-l} + \epsilon_{r,t},$$

where $r_{m,t}$ is the market return, $x_t$ is the signed dollar volume of trading (positive values for net buying and negative values for net selling), and $r_t$ is the bitcoin return. We estimate reduce-form VAR and use the reduced-form error covariances to recover the SVAR parameters.

$$r_{m,t}^* = a_0^* + \sum_{l=1}^{5} a_{1,l}^* r_{m,t-l} + \sum_{l=1}^{5} a_{2,l}^* x_{t-l} + \sum_{l=1}^{5} a_{3,l}^* r_{r,t-l} + e_{r_m,t}$$

$$x_t^* = b_0^* + \sum_{l=0}^{5} b_{1,l}^* r_{m,t-l} + \sum_{l=1}^{5} b_{2,l}^* x_{t-l} + \sum_{l=1}^{5} b_{3,l}^* r_{r,t-l} + e_{x,t} \tag{C15}$$

$$r_t^* = c_0^* + \sum_{l=0}^{5} c_{1,l}^* r_{m,t-l} + \sum_{l=1}^{5} c_{2,l}^* x_{t-l} + \sum_{l=1}^{5} c_{3,l}^* r_{r,t-l} + e_{r,t},$$

Hence, we have:

$$e_{r_m,t} = \epsilon_{r_m,t}$$
$$e_{x,t} = \epsilon_{x,t} + b_{1,0} \epsilon_{r_m,t} = b_{1,0} e_{r_m,t} + \epsilon_{x,t} \tag{C16}$$
$$e_{r,t} = \epsilon_{r,t} + (c_{1,0} + c_{2,0} b_{1,0}) \epsilon_{r_m,t} + c_{2,0} \epsilon_{x,t} = c_{1,0} e_{r_m,t} + c_{2,0} e_{x,t} + \epsilon_{r,t},$$

and:

$$\sigma_{\epsilon_{r_m}}^2 = \sigma_{e_{r_m}}^2$$
$$\sigma_{\epsilon_x}^2 = \sigma_{e_x}^2 - b_{1,0}^2 \sigma_{e_{r_m}}^2 \tag{C17}$$
$$\sigma_{\epsilon_r}^2 = \sigma_{e_r}^2 - (c_{1,0}^2 + 2 c_{1,0} c_{2,0} b_{1,0}) \sigma_{e_{r_m}}^2 - c_{2,0}^2 \sigma_{e_x}^2.$$

Rewrite the reduced-form residuals as linear functions of the structural-model residuals:

$$e_{r_m,t} = \epsilon_{r_m,t}$$
$$e_{x,t} = \epsilon_{x,t} + b_{1,0} \epsilon_{r_m,t}$$
$$e_{r,t} = \epsilon_{r,t} + (c_{1,0} + c_{2,0} b_{1,0}) \epsilon_{r_m,t} + c_{2,0} \epsilon_{x,t}$$

- Estimate $b_{1,0}$ by regressing the reduced-form innovation $e_{x,t}$ on $\epsilon_{r_m,t}$

- Estimate $c_{1,0}$ and $c_{2,0}$ by regressing the reduced-form innovation $e_{r,t}$ on $\epsilon_{r_m,t}$ and $\epsilon_{x,t}$

The structural-model residuals are contemporaneously uncorrelated by construction. The reduced-form residuals are contemporaneously correlated, and the contemporaneous correlation can be used to infer the structural-model residuals. From the estimated parameters $b_{1,0}$, $c_{1,0}$, and $c_{2,0}$, and the estimated variances of the reduced-form residuals $\sigma^2_{\epsilon_{r_m}}$, $\sigma^2_{\epsilon_x}$, and $\sigma^2_{\epsilon_r}$, we obtain estimates of the variances of the structural model shocks by:

- A structural shock to market returns $[\epsilon_{r_m,t}, \epsilon_{x,t}, \epsilon_{r,t}] = [1, 0, 0]'$ has a reduced-form equivalent $[\epsilon_{r_m,t}, \epsilon_{x,t}, \epsilon_{r,t}] = [0, b_{1,0}, (c_{1,0} + c_{2,0}b_{1,0})]'$.
- A structural shock to trading $[\epsilon_{r_m,t}, \epsilon_{x,t}, \epsilon_{r,t}] = [0, 1, 0]'$ has a reduced-form equivalent $[\epsilon_{r_m,t}, \epsilon_{x,t}, \epsilon_{r,t}] = [0, 1, c_{2,0}]'$.
- A structural shock to the stock returns $[\epsilon_{r_m,t}, \epsilon_{x,t}, \epsilon_{r,t}] = [0, 0, 1]'$ has a reduced-form equivalent $[\epsilon_{r_m,t}, \epsilon_{x,t}, \epsilon_{r,t}] = [0, 0, 1]'$.

The cumulative return response to each of these shocks gives estimates of $\theta_{r_m}$, $\theta_x$, and $\theta_r$, respectively.

$$r_t = \mu + \theta_{r_m}\epsilon_{r_m,t} + \theta_x\epsilon_{x,t} + \theta_r\epsilon_{r,t} + \Delta s_t \tag{C18}$$

The final step is to combine the estimated $\theta_{r_m}$, $\theta_x$, and $\theta_r$ with the estimated variances of the reduced-form residuals $\sigma^2_{\epsilon_{r_m}}$, $\sigma^2_{\epsilon_x}$, and $\sigma^2_{\epsilon_r}$ to get the variance components and component shares.

## D  SVAR and Information shares

Figures D1 D2 D3 D4 show the value over time of market information, public information, private information, and noise shares for each stock, with darker colors taking on larger values. The results show that public information has the largest share, market information has the second largest share, noise has the third largest share, and private information has the smallest share.

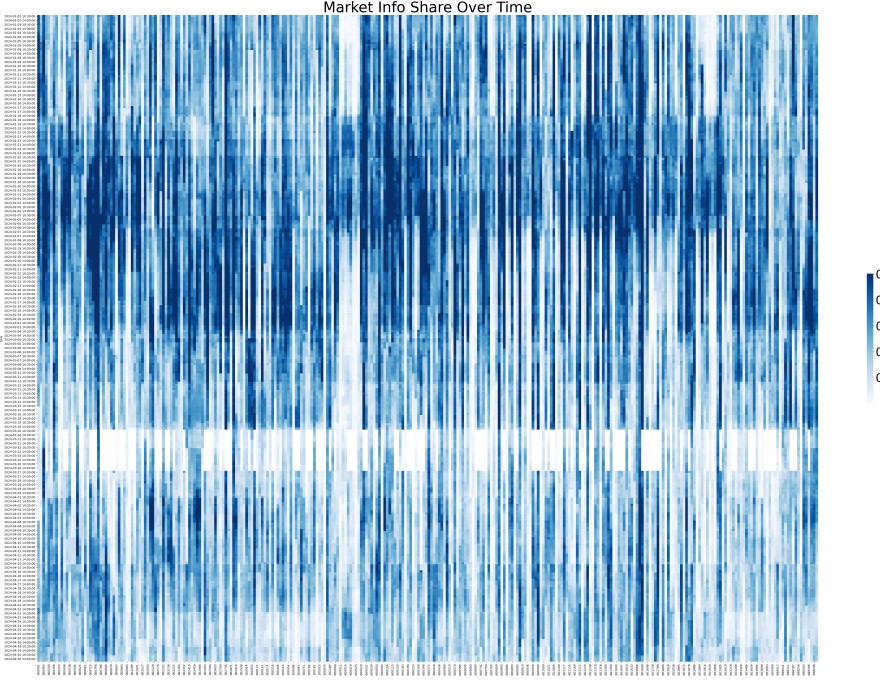

Figure D1: Market info share

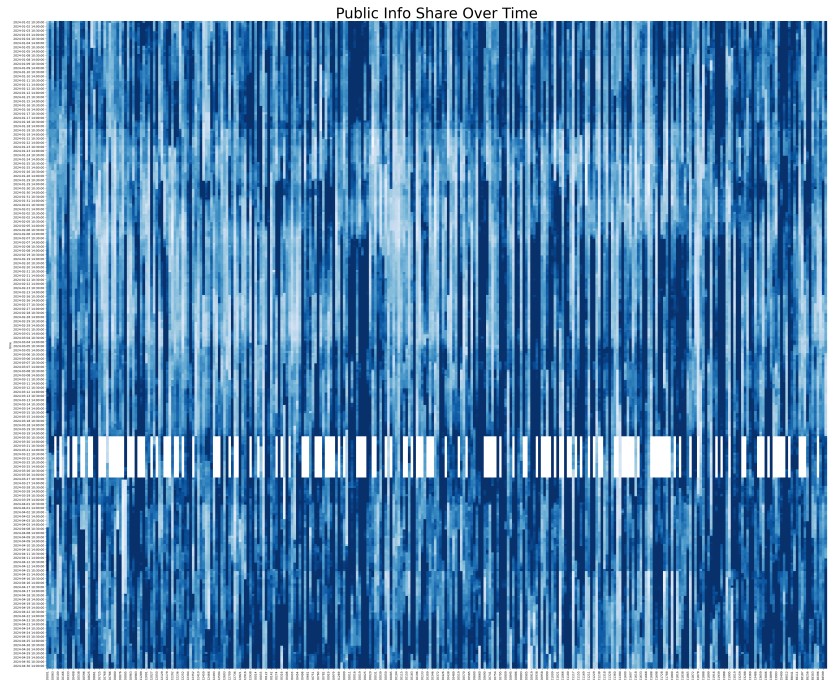

Figure D2: Public info share

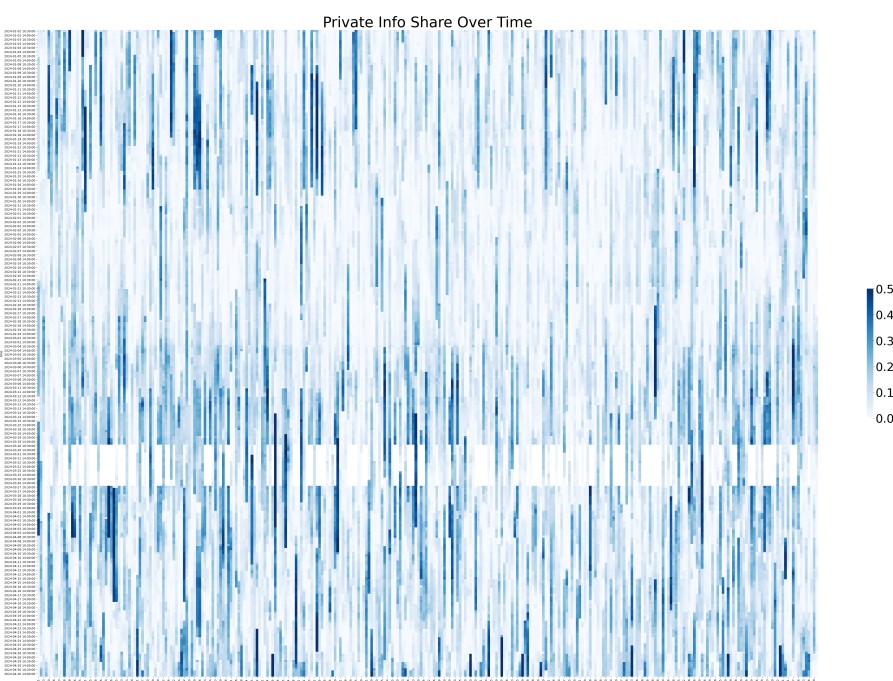

Figure D3: Private info share

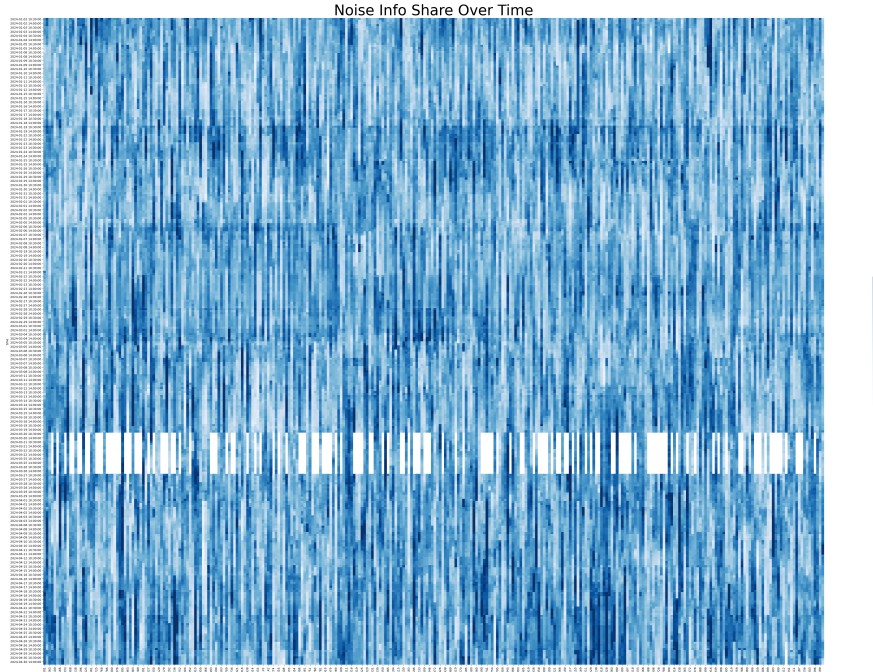

Figure D4: Noise info share

# E    PORTFOLIO ROBUSTNESS

First, we change the time series scale from 60 to 50 and 40 hours, and construct the portfolios based on the market information of SVAR and public and market information, respectively, and the results are shown in FigureE1 Our results show that the portfolios still have positive returns after changing the time series scale, and the addition of the SVAR module can significantly improve the portfolio performance compared to the DiT and LSTM models. In the model with a time series scale of 40 hours, based on public and market information, DiT-LSTM-SVAR has an annual return of 128.48% with a sharpe ratio of 1.20. Based on public information DiT-LSTM-SVAR has an annual return of 124.00% with a sharpe ratio of 0.91. 0.91. In the model with a time series scale of 50 hours, based on public and market information, the annual return of DiT-LSTM-SVAR is 99.67%, and the sharpe ratio is 0.83. Based on public information, the annual return of DiT-LSTM-SVAR is 170.73%, and the sharpe ratio is 1.20. Based on public information, DiT-LSTM-SVAR has an annual return of 170.73% and a sharpe ratio of 1.15.

Second, we change the portfolio strategy from hourly decision making to daily decision making, every two days decision making and every three days decision making and construct the portfolios based on the market information from SVAR and public and market information, respectively, and the results are shown in the FigureE2. Our results show that the portfolios still have positive returns after changing the investment strategy, and the addition of the SVAR module can significantly improve the portfolio performance compared to the DiT and LSTM models. In the model of daily decision making, based on public and market information, the annual return of DiT-LSTM-SVAR is 145.43% with a sharpe ratio of 1.15. Based on public information DiT-LSTM-SVAR has an annual return of 115.39% with a sharpe ratio of 0.87. In the model of bi-daily decision making, based on public and market information, the annual return is 115.39% with a sharpe ratio of 0.87. In the model of daily decision making, based on public and market information, the annual return is 145.43% with a sharpe ratio of 1.15. In the model of two-day decision making, based on public and market information, the annual return of DiT-LSTM-SVAR is 102.67%, and the sharpe ratio is 0.92. In the model of three-day decision making, based on public information, the annual return of DiT-LSTM-SVAR is 96.24%, and the sharpe ratio is 0.76. In the model of three-day

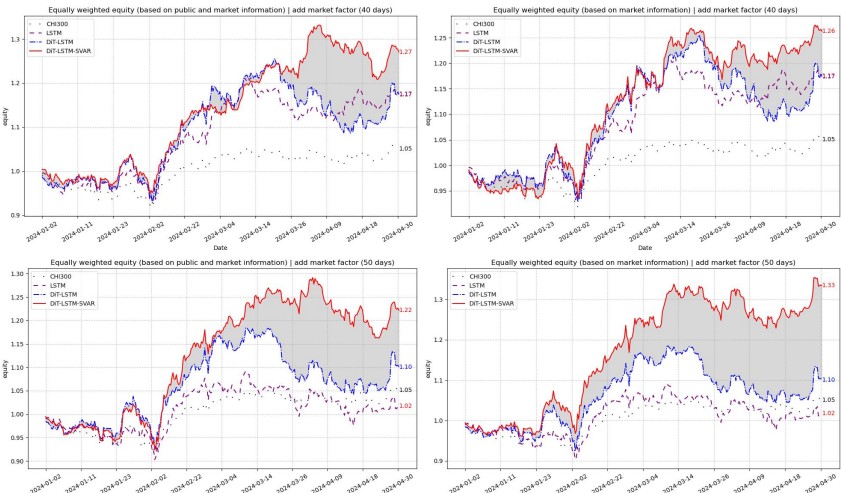

Figure E1: Change time series scale

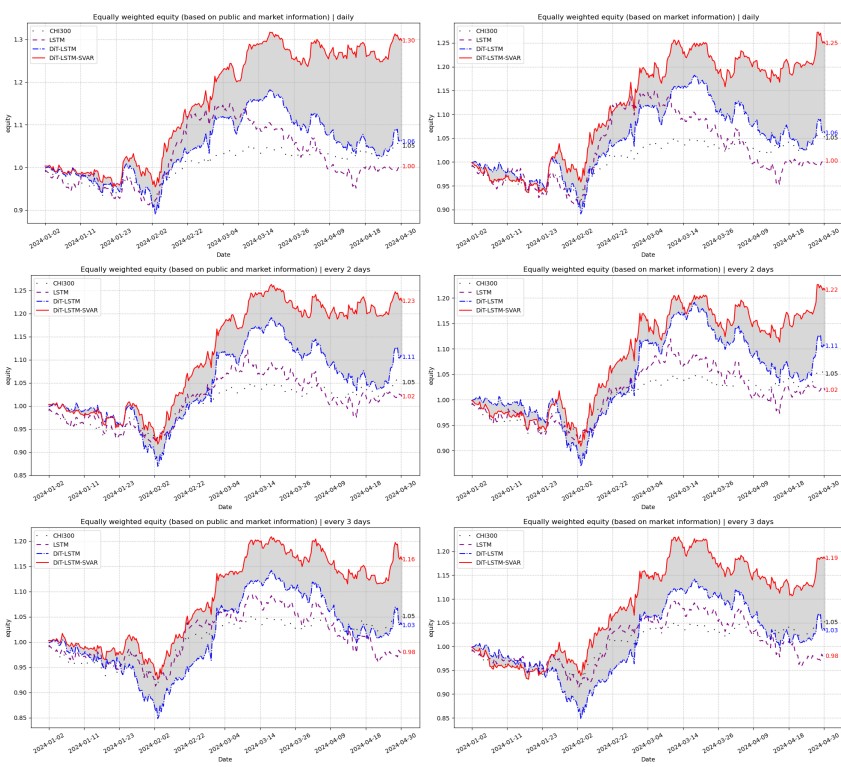

Figure E2: Change portfolio strategy

decision making, based on public information, the annual return is 1.15. In the model of three-day decision making, based on public information, the annual return is 1.15. In the model of decision making based on public information and market information, the annual return of DiT-LSTM-SVAR is 68.63%, and the sharpe ratio is 0.72. The annual return of DiT-LSTM-SVAR based on public information is 79.83%, and the sharpe ratio is 0.73.

Third, we have considered the issue of transaction fees in the market. Our following experiment is at a transaction cost of one thousandth with market factors, and the results are shown in the FigureE3. Our result shows that at a transaction cost of one thousandth,

the investment performance of the basic LSTM model showed a significant decline after incorporating market features. However, after running our combination model DiT-LSTM-SVAR, this downward trend was fundamentally reversed, and the cumulative return finally rise to 1.07. This further proves the effectiveness of our model.

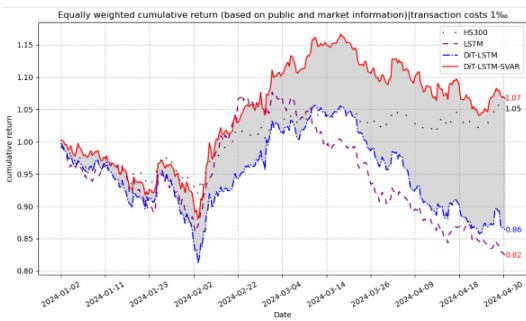

Figure E3: Portfolio performance under 1‰ transaction costs

## F Model Comparison

First, we test the standard Transformer model under the enhanced approach. To balance efficiency and computational accuracy, our Transformer model includes a two-layer single-head attention mechanism and a fully connected layer output with a clean design similar to the LSTM, with other parameter settings similar to the LSTM. The results are shown in the FigureF1 and TableF1. Our results show that transformer models can also achieve good results in time prediction. And our enhancement method has indeed brought performance improvement when applied to transformer models, further verifying the effectiveness of the method. In the model of transformer-based methods, the annual return of Transformer is 14.32% with a sharpe ratio of 0.51. Since applying our enhancement method, the annual return of DiT-Transformer-SVAR can reach 29.80% with a sharpe ratio of 1.02 at most, which is a significant improvement. Although the performance of the basic transformer model is slightly better than that of the basic LSTM model, the application of our enhancement method significantly improves the performance of the LSTM model, surpassing the enhanced Transformer model.

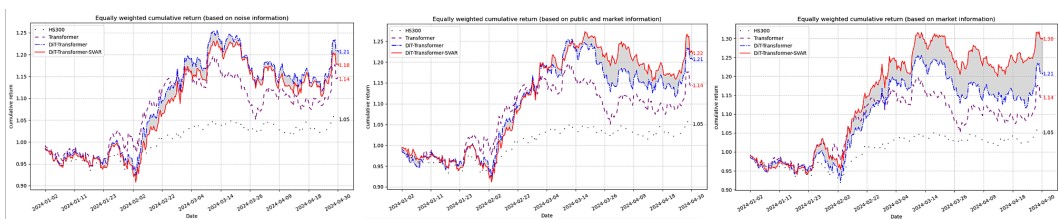

Figure F1: Portfolio performance of transformer-based models

Table F1: Portfolio performance of transformer-based models

| Model | Cumulative Return | Annual Return | MDD | Annual STD | Sortino | VaR95 | Sharpe |
|---|---|---|---|---|---|---|---|
| CHI300 | 5.05% | 18.54% | -7.39% | 16.04% | 0.52 | -0.86% | 0.31 |
| Transformer | 14.32% | 58.74% | -'12.28% | 29.62% | 0.97 | -1.26% | 0.51 |
| DiT-Transformer | 20.58% | 90.79% | -11.08% | 30.58% | 1.18 | -1.58% | 0.67 |
| DiT-Transformer-SVAR(no market factor,MP) | 22.07% | 99.03% | -9.34% | 27.72% | 1.35 | -1.31% | 0.80 |
| DiT-Transformer-SVAR(no market factor, M) | 29.80% | 146.03% | -8.90% | 29.17% | 1.94 | -1.28% | 1.02 |
| DiT-Transformer-SVAR(no market factor, N) | 17.52% | 74.57% | -10.40% | 30.15% | 1.01 | -1.59% | 0.58 |

Second, we change the generation method to GAN to verify the robustness of our model. In order to reflect the concept of model simplicity, our GAN model is designed to be as simple as the LSTM model. The generator consists of a layer of LSTM, a layer of GRU, and a

layer of linear layer; the discriminator consists of a layer of Conv1d and a layer of linear layer. Other parameters such as the learning rate design are the same as the LSTM model. The results are shown in the FigureF2 and TableF2. Our results show that the GAN model can also effectively improve the portfolio performance of the LSTM model, specifically, the annual return increased from 12.58% to 20.55%, and the sharpe ratio increased from 0.48 to 0.78. Furthermore, after applying the SVAR method, the annual return can be increased to 27.14% and the sharpe ratio can be increased to 1.10 at most, demonstrating its unique advantages in generative adversarial networks. This result further verifies the feasibility of our framework. Also, it is worth noting that noise-based screening methods may have a negative impact on investment performance. However, this contribution to the investment portfolio is not as significant as the DiT model.

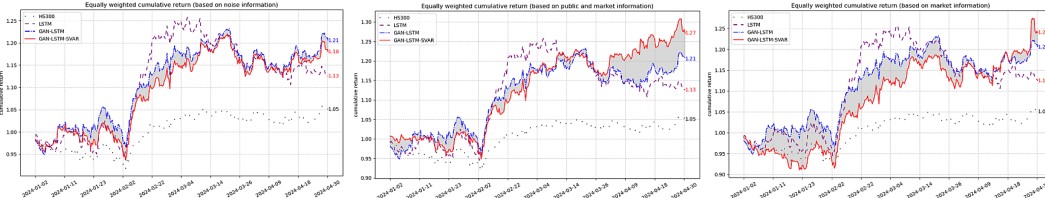

Figure F2: Portfolio performance of GAN-based models

Table F2: Portfolio performance of GAN-based models

| Model | Cumulative Return | Annual Return | MDD | Annual STD | Sortino | VaR95 | Sharpe |
|---|---|---|---|---|---|---|---|
| CHI300 | 5.05% | 18.54% | -7.39% | 16.04% | 0.52 | -0.86% | 0.31 |
| LSTM | 12.58% | 50.55% | -12.04% | 26.21% | 0.82 | -1.25% | 0.48 |
| GAN-LSTM | 20.55% | 90.63% | -8.54% | 26.29% | 1.38 | -1.25% | 0.78 |
| GAN-LSTM-SVAR(no market factor,MP) | 27.14% | 129.09% | -6.82% | 24.71% | 1.77 | -1.10% | 1.10 |
| GAN-LSTM-SVAR(no market factor, M) | 24.03% | 110.32% | -7.86% | 28.22% | 1.63 | -1.20% | 0.85 |
| GAN-LSTM-SVAR(no market factor, N) | 17.96% | 76.87% | -8.27% | 25.86% | 1.23 | -1.26% | 0.69 |

Third, we provide examples of using another financial method, CVaR, to enhance the model. Conditional Value at Risk (CVaR), also known as Expected Shortfall, is a risk measure used in finance to assess the potential losses of an investment portfolio. It provides an estimate of the expected loss beyond a given Value at Risk (VaR) threshold, offering a more comprehensive view of tail risk. The results are shown in the FigureF3. Our results show that the CVaR method not only failed to enhance the existing model, but also reduced the cumulative return of the model from 1.23 to 1.22. We can see that the effect of using CVaR method is not as good as SVAR. The reason might be that CVaR primarily focuses on the average tail risk under normal market conditions, which is the expected loss given that losses exceed the VaR threshold. It relies on regular market volatility and assumes that historical data can represent future risks. However, this method may not fully capture the complex dynamic relationships in the market. In contrast, the SVAR method incorporates structural information from economic theory, considering the interrelationships and dynamic changes between variables to assess risk. This allows SVAR to better capture the interactions and potential transmission effects between different market variables, providing a more accurate risk assessment. Therefore, in primary markets, SVAR can more comprehensively reflect the risk landscape, making it more effective than CVaR.

# G    LSTM

**LSTM**. Long Short-Term Memory (LSTM) networks are a special kind of Recurrent Neural Network (RNN) capable of learning long-term dependencies. LSTM networks are designed to address the vanishing and exploding gradient problems commonly found in traditional RNNs. They do this through the use of several gates that regulate information flow, making them more effective at preserving the context over long sequences. The main components of an LSTM cell include:

- **Cell State:** Acts as a conveyor belt, running through the chain with minimal interaction.

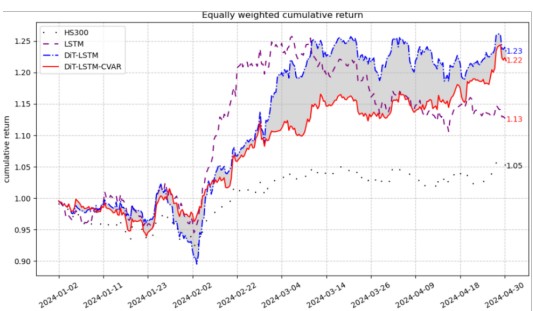

Figure F3: Portfolio performance of CVaR-based models

- **Input Gate:** Decides the amount of new information to be added to the cell state.
- **Forget Gate:** Determines what information is discarded from the cell state.
- **Output Gate:** Controls the output flow from the cell state to the rest of the network.

**GRU**. GRUs simplify the LSTM architecture by combining the forget and input gates into a single "update gate." Another gate, the "reset gate," is used to determine how much past information to forget. The GRU's architecture makes it easier to modify and often faster to train, without a significant drop in performance compared to LSTMs. The main components of a GRU are:

- **Update Gate:** Controls the extent to which a GRU unit updates its activation, or content state.
- **Reset Gate:** Decides how much of the past information to forget.

