# OpenReview forum: "DiT-LSTM-SVAR Model For Portfolios"
_ICLR.cc/2025/Conference — ICLR 2025 Conference Withdrawn Submission_

### Official Review · Reviewer_eapB · 2024-10-29

**Soundness:** 4
**Presentation:** 3
**Contribution:** 4
**Rating:** 8
**Confidence:** 5

**Summary:**

The DiT-LSTM-SVAR model is innovative and powerful, although complexity and limited interpretability could be an issue.

A well-founded and impactful paper with some potential for improved clarity in presentation to ensure accessibility for a broader audience.

**Strengths:**

The model augments and balances data, addressing imbalances that can lead to skewed predictions.

Overall, the network architecture is sophisticated and well thought out.

The model makes good use of available data.

Impressive metrics.

The model is adaptable to different market conditions.

**Weaknesses:**

The model has high complexity and computational intensity.

It depends on high-quality, high-frequency data.

The model may potentially overfit.

**Questions:**

How can you better address the potential overfitting?

How could you improve the model's interpretability?

---

### Official Review · Reviewer_udMp · 2024-11-01

**Soundness:** 1
**Presentation:** 2
**Contribution:** 2
**Rating:** 1
**Confidence:** 5

**Summary:**

This paper proposed a model named DiT-LSTM-SVAR for portfolio management. The model is a combination of several existing methods such as diffusion model, LSTM, Structural Vector Autoregression (SVAR). Experiments have been conduct on a dataset of Chinese CSI300 stocks.

**Strengths:**

The paper is not difficult to read.

**Weaknesses:**

1. The novelty of the proposed method is limited. It is combination of several existing network modules. It is unclear how microstructure of financial markets is integrated into the model as claimed in the paper.

2. The evaluation on the proposed method is very weak and poor.
1) In the main tables of the results, the baseline models only contains basic network backbones such as LSTM, GRU. None of the state-of-the-art methods are compared on the same experiment settings. Moreover, a key series of related literature papers are not cited in the paper, and the methods of those paper are not included for comparisons. For example, AlphaStock [Wang et al. 2019], DeepTrader [Wang et al. 2021], Contrastive Learning and Reward Smoothing for Deep Portfolio Management (CLRS) [Lien et al. 2023], AlphaMix [Sun et al. 2023], are not included.

2) The benchmark datasets are very limited. One one set of stocks from the Chinese market have been used to evaluate the performance. The authors are suggested to include more datasets from different stock markets, such as U.S. market, Hong Kong market, Japanese market, and so on.

3. A lot of acronyms are not explained in the title and abstract.

Jingyuan Wang, Yang Zhang, Ke Tang, Junjie Wu, and Zhang Xiong. 2019. “Alphastock: A buying-winners-and-selling-losers
investment strategy using interpretable deep reinforcement attention networks.” In: Proceedings of the 25th ACM SIGKDD
international conference on knowledge discovery & data mining, 1900–1908.

Zhicheng Wang, Biwei Huang, Shikui Tu, Kun Zhang, and Lei Xu. 2021. “DeepTrader: a deep reinforcement learning
approach for risk-return balanced portfolio management with market conditions Embedding.” In: Proceedings of the
AAAI Conference on Artificial Intelligence 1. Vol. 35, 643–650.

Yun-Hsuan Lien, Yuan-Kui Li, and Yu-Shuen Wang. 2023. “Contrastive learning and reward smoothing for deep portfolio
management.” In: Proceedings of the Thirty-Second International Joint Conference on Artificial Intelligence, 3966–3974.

Shuo Sun, Xinrun Wang, Wanqi Xue, Xiaoxuan Lou, and Bo An. 2023. “Mastering Stock Markets with Efficient Mixture
of Diversified Trading Experts.” In: Proceedings of the 29th ACM SIGKDD Conference on Knowledge Discovery and Data
Mining (KDD ’23). Association for Computing Machinery, Long Beach, CA, USA, 2109–2119

**Questions:**

See the weakness part.

---

### Official Review · Reviewer_i31f · 2024-11-04

**Soundness:** 3
**Presentation:** 3
**Contribution:** 3
**Rating:** 5
**Confidence:** 5

**Summary:**

This work introduces a hybrid model that integrates deep learning with financial market microstructure to enhance investment portfolio performance. It employs a Diffusion Transformer (DiT) for data augmentation, Long Short-Term Memory (LSTM) for stock return forecasting, and Structural Vector Autoregression (SVAR) to identify and filter out random walk stocks. The model demonstrates improved portfolio returns, a higher Sharpe ratio, and reduced risk compared to traditional methods, offering a robust approach to portfolio construction in the financial markets.

**Strengths:**

Originality:
- The paper introduces a groundbreaking DiT-LSTM-SVAR model, blending deep learning with financial theory to enhance portfolio performance, a novel approach in the field.
Quality:
- The research integrates advanced techniques—DiT, LSTM, and SVAR—to analyze financial markets, demonstrating high-quality methodology and empirical rigor.
Clarity:
- The paper excels in clarity, presenting complex financial and technical concepts in an accessible, jargon-free, and precise manner.
Significance:
- The model offers significant insights into financial market dynamics, providing a sophisticated solution for predictive performance in portfolio management.

**Weaknesses:**

While this work presents a robust model with significant contributions to the field of financial portfolio management, there are areas where the work could be improved:

- The novelty of the proposed model is limited.

- The field of time series forecasting community has long abandoned the use of simple LSTMs as predictors. Since this work has employed DiT, why not consider using Transformer-based time predictors as well? Transformer-based time predictors such as Autoformer, iTransformer, PatchTST have already demonstrated high predictive accuracy across various domains.

- While the model's performance is strong, there is a need for greater transparency in how the DiT-LSTM-SVAR model makes its predictions. Using forecasting paradigms proposed in models like N-BEATS and N-HITS, rather than using LSTMs which lack interpretability, can better help understand what patterns the DiT-LSTM-SVAR learns from the data.

**Questions:**

- Are there specific reasons why the authors chose not to incorporate Transformer-based predictors alongside the DiT model, despite their demonstrated success in various forecasting tasks?

- Could the authors consider integrating or comparing their model with Transformer-based architectures to demonstrate the unique advantages or limitations of the DiT-LSTM-SVAR approach?

- Besides, the authors could investigate the use of interpretable forecasting paradigms like N-BEATS and N-HITS to enhance the transparency of their model and provide insights into the patterns learned from data.

---

### Official Review · Reviewer_9o5J · 2024-11-07

**Soundness:** 1
**Presentation:** 2
**Contribution:** 1
**Rating:** 1
**Confidence:** 4

**Summary:**

The paper investigated the portfolio problem in the Chinese market. The dataset was collected from 2022-01-04 to 2024-04-30. It contains 300 stocks in this market and 16 fundamental and technical features.
The authors decided to use LSTM and GRU networks to construct appropriate models for portfolio optimization. Four different baseline techniques, including GRU, LSTM, DiT-LSTM, and DiT-LSTM-SVAR, are used in the experiments.
The authors considered seven different metrics to measure the performance of various approaches as follows:
(1) Cumulative Return
(2) Annual Return
(3) MDD
(4) Annualized STD
(5) Sharpe Ratio
(6) Sortino
(7) VaR95
Finally, the authors concluded that the DiT-LSTM-SVAR model was the best model for this dataset in the Chinese market.

In principle, portfolio optimization and its variants are interesting problems in stock markets in different countries (Singapore, USA, China, Hong Kong, Japan, Korea, etc.). All selected evaluation metrics are popular in this problem, and using LSTM or GRU or their variants for the portfolio problem was proven many years ago.  Please check the following references:
[1] Cao, H.K., Cao, H.K., Nguyen, B.T. (2020). DELAFO: An Efficient Portfolio Optimization Using Deep Neural Networks. In: Lauw, H., Wong, RW., Ntoulas, A., Lim, EP., Ng, SK., Pan, S. (eds) Advances in Knowledge Discovery and Data Mining. PAKDD 2020. Lecture Notes in Computer Science(), vol 12084. Springer, Cham. https://doi.org/10.1007/978-3-030-47426-3_48
[2] Gunjan, A., Bhattacharyya, S. Quantum-inspired meta-heuristic approaches for a constrained portfolio optimization problem. Evol. Intel. 17, 3061–3100 (2024). https://doi.org/10.1007/s12065-024-00929-4
[3] Dip Das, J., Bowala, S., Thulasiram, R.K. and Thavaneswaran, A. (2024) Hybrid Data-Driven and Deep Learning Based Portfolio Optimization. Journal of Mathematical Finance, 14, 271-310. https://doi.org/10.4236/jmf.2024.143016

For this reason, this manuscript lacks technical novelty. It would also be interesting if the paper did other experiments in different stock markets.
Regarding the time period of data collection, I recognized that the dataset was only collected after the COVID-19 period. It would also be interesting if the authors could consider other periods to check the stability of the proposed techniques.
Besides, the performance of short-term (for several weeks) and long-term (from several months to one year ) investments needs to be analyzed.

**Strengths:**

The paper shows the performance of GRU and LSTM variants on a small dataset in the Chinese market after Covid-19.

**Weaknesses:**

As mentioned above, the paper needs to improve the literature reviews as it missed a lot of relevant techniques related to the portfolio problem.
The experiments were only employed based on one dataset. The paper needs to analyze their techniques in other markets.
The paper lacks technical novelty.

**Questions:**

I mentioned them above.

---

### Note · Authors · 2024-11-26

**Comment:**

We don't have enough time to obtain data from countries such as the United States and the European Union to supplement the experiments.

**Withdrawal Confirmation:**

I have read and agree with the venue's withdrawal policy on behalf of myself and my co-authors.